# Logarithmic Regret for Unconstrained Submodular Maximization Stochastic Bandit

**Julien Zhou**                                        JULIEN.ZHOU@INRIA.FR
*Criteo AI Lab, Paris, France*
*Univ. Grenoble Alpes, Inria, CNRS, Grenoble INP, LJK, 38000 Grenoble, France*
**Pierre Gaillard**                                    PIERRE.GAILLARD@INRIA.FR
*Univ. Grenoble Alpes, Inria, CNRS, Grenoble INP, LJK, 38000 Grenoble, France*
**Thibaud Rahier**                                     T.RAHIER@CRITEO.COM
*Criteo AI Lab, Paris, France*
**Julyan Arbel**                                       JULYAN.ARBEL@INRIA.FR
*Univ. Grenoble Alpes, Inria, CNRS, Grenoble INP, LJK, 38000 Grenoble, France*

**Editors:** Gautam Kamath and Po-Ling Loh

## Abstract

We address the *online unconstrained submodular maximization problem* (Online USM), in a setting with *stochastic bandit feedback*. In this framework, a decision-maker receives noisy rewards from a non monotone submodular function taking values in a known bounded interval. This paper proposes *Double-Greedy - Explore-then-Commit* (DG-ETC), adapting the Double-Greedy approach from the offline and online full-information settings. DG-ETC satisfies a $O(d \log(dT))$ problem-dependent upper bound for the $1/2$-approximate pseudo-regret, as well as a $O(dT^{2/3} \log(dT)^{1/3})$ problem-free one at the same time, outperforming existing approaches. In particular, we introduce a problem-dependent notion of hardness characterizing the transition between logarithmic and polynomial regime for the upper bounds.

**Keywords:** Submodular maximization; combinatorial optimization; stochastic bandits; logarithmic regret.

## 1. Introduction

### 1.1. Context and problem formulation

Several real-world settings can be cast as combinatorial optimization problems over a finite set. Without some assumptions on the utility function to be maximized and/or the constraints to be satisfied, such problems cannot be solved in polynomial time. In practice, different types of assumptions and constraints can be introduced to make these problems manageable, even approximately. One can, for example, assume the utility to be linear, but in some cases even this already strong assumption can be helpless to make the problem easier.

This paper focuses on the cases where we maximize a submodular set-function, meaning that it satisfies a "diminishing marginal gains" property. We consider the unconstrained setting, where the whole combinatorial super-set is available and the utlility may be nonmonotone (if we know that it is monotone, the solution is straightforward, being either the full or the empty set). We also place ourselves in a stochastic (combinatorial) bandit setting, where a *decision-maker / player* chooses different sets in sequential rounds, and receives noisy rewards. In this framework, a classic challenge is to balance exploration and exploitation, but the problem of managing the combinatorial complexity of the action set is stacked over it. In particular, a good strategy should efficiently leverage the underlying structure of the reward – submodularity in this case – by monitoring relevant quantities.

**Problem formulation and assumptions.** We consider a finite set of $d \in \mathbb{N}^*$ *items* $\mathcal{D}$. The player has access to *actions* from the whole superset $\mathcal{P}(\mathcal{D})$ and plays for an *horizon* of $T \in \mathbb{N}^*$ rounds. The player receives noisy *rewards* from a *non monotone submodular* set-function $f : \mathcal{P}(\mathcal{D}) \to [0, c]$ with $c > 0$. At each round $t \in [T] = \{1, \ldots, T\}$, the player chooses an action $A_t \in \mathcal{P}(\mathcal{D})$ and receives

$$Z_t = f(A_t) + \eta_t \, , \tag{1}$$

where $\eta_t$ is a random variable. Let $\sigma > 0$ (known), we assume that $\eta_t$ is $\sigma^2$-sub-Gaussian conditionally to the past (including the possibly random process generating $A_t$).

The algorithms that we study in this paper all consider items sequentially. For convenience, we identify $\mathcal{D}$ with $[d]$ and assume an arbitrary ordering, but a player with prior knowledge could try to optimize over permutations.

**$1/2$-Approximate pseudo-regret minimization.** The objective of the player is to maximize its cumulative rewards over the $T$ rounds. As it is common in the bandit literature, we look instead at a pseudo-regret, neglecting the contribution of the noise $(\eta_t)_{t \in [T]}$. Besides, rather than looking at the exact pseudo-regret, we minimize an $1/2$-approximation defined as

$$R_T = \sum_{t=1}^{T} \left[ \frac{1}{2} f(A^*) - f(A_t) \right] , \tag{2}$$

where $A^* \in \arg\max_{A \subseteq \mathcal{D}} \left\{ f(A) \right\} .$

Considering approximate regrets is usual in settings where we have access to an *oracle* solving the offline optimization approximately (Chen et al., 2013). In our framework, the $1/2$ factor comes from the impossibility of solving the offline unconstrained submodular maximization problem (USM), with a competitive ratio better than $1/2$, using a polynomial number of calls (Feige et al., 2011).

In the following, if not specified, the expressions "pseudo-regret" or just "regret" refer to the $1/2$-*approximate pseudo-regret*.

## 1.2. Contributions

We propose a novel algorithm *Double-Greedy - Explore-then-commit* (`DG-ETC`) for the online unconstrained submodular maximization problem (Online USM), with stochastic bandit feedback (Section 3). We introduce a new notion of *hardness* for this problem (Section 4.1), and prove that `DG-ETC` satisfies both a logarithmic problem-dependent (hardness-dependent) upper bound for the $1/2$-approximate pseudo-regret, as well as a worst-case $O(dT^{2/3} \log(dT)^{1/3})$ upper bound (Sections 4.2 and 5). These bounds are satisfied both with high-probability and in expectation (Theorem 2), and rely on the stationarity of the stochastic setting. Asymptotically, `DG-ETC` allocates a logarithmic, hardness-dependent, number of rounds to the design of a strategy that compensates the randomness errors with per-round negative losses (therefore, with gains). In practice, `DG-ETC` exploits the looseness of the $1/2$-approximation ratio in non-adversarial cases, and we argue that this kind of strategy could also be applied to other settings involving approximations.

### 1.3. Related works

In this section, we mention the closest related works, concerning combinatorial bandits, offline (unconstrained) submodular maximization, as well as its online and bandit versions. Supplementary discussions with other lines of work can be found Appendix A (offline and online minimization, constrained maximization, and other online maximization problems).

**Combinatorial bandits.** The recent monograph by Lattimore and Szepesvári (2020) makes an extensive study of bandit problems. We are more particularly interested in settings where the action space is combinatorial and too big to be explored in its entirety. Chen et al. (2013) (extended by Chen et al. (2016)) introduces the stochastic semi-bandit framework, and derives results for approximate pseudo-regrets and smooth, monotone aggregation functions. When the aggregation is linear, the leading factors in the regret upper bounds have been refined in several subsequent works (Kveton et al., 2015; Degenne and Perchet, 2016; Perrault et al., 2020; Zhou et al., 2024). A matching adversarial semi-bandit setting has also been explored (Ito, 2021; Neu and Valko, 2014). While the player gets one feedback per chosen item in the semi-bandit setting, the full-bandit (or just "bandit") setting with a single feedback per action is more challenging. If the aggregation remains linear, one could see the problem as a linear bandit and use the corresponding methods, as long as the offline problem can be solved (Abbasi-Yadkori et al., 2011; Bubeck et al., 2012). However, Considering a nonlinear aggregation function with a full-bandit feedback remains challenging without further assumptions (Han et al., 2021).

**Unconstrained sumbodular maximization (USM).** Several systems can be modeled with a submodular structure in various fields, including economics, game theory and combinatorial optimization. As it shares properties similar to both convexity and concavity in continuous optimization (Lovász, 1983), both viewpoints are of interest. The monograph by Bach (2013) details various cases where submodular set-functions appear and highlights the parallels between submodular minimization and convex optimization. While minimization can be solved in polynomial time, maximization is more challenging and can in general only be solved approximately (Feige et al., 2011). A $(1 - 1/e)$-approximation is possible in the cardinally-constrained monotone case (Nemhauser et al., 1978), but the unconstrained non monotone setting can only be solved up to a $1/2$ approximation ratio (Feige et al., 2011). In particular, Buchbinder et al. (2012) provides a linear-time approach reaching this ratio, closing the gap between upper and lower bounds.

**Online USM with full-information and bandit feedback.** Following the results from Buchbinder et al. (2014), Roughgarden and Wang (2018) studies the particular case of non monotone, unconstrained maximization in the online adversarial full-information setting and provides an algorithm satisfying a $O(d\sqrt{T})$ regret upper bound. Harvey et al. (2020) manages to gain a $\sqrt{d}$ factor by using tools related to online dual averaging and Blackwell approachability. Fourati et al. (2023) considers a stochastic bandit setting, and proposes an Explore-then-Commit type algorithm satisfying a $O(dT^{2/3}\log(T)^{1/3})$ regret upper bound. However, Niazadeh et al. (2021) claims a similar $O(dT^{2/3})$ in an adversarial bandit setting. As the latter framework seems significantly more difficult, one may reasonably wonder if better guarantees can be satisfied in the stochastic setting. We answer this question positively and propose an algorithm satisfying both logarithmic problem-dependent and $O(d(T\log(dT))^{2/3})$ problem-free upper bounds.

## 2. Preliminary

In this section, we introduce submodularity, and remind the approach of the Double-Greedy (`DG`) algorithm (Buchbinder et al., 2012) on which our `DG-ETC` is based.

### 2.1. Submodularity

Submodularity is a "diminishing marginal gains" property, it is formally defined as follows.

**Definition 1 (Submodularity)**   *Let $\mathcal{D}$ be a finite set and $c > 0$. A set-function $f : \mathcal{P}(\mathcal{D}) \to [0, c]$ is said to be (bounded)* submodular *if, equivalently,*
- *For all $A \subseteq B \subseteq \mathcal{D}$ and $i \in \mathcal{D}$,     $f(B \cup \{i\}) - f(B) \leq f(A \cup \{i\}) - f(A)$ ;*
- *For all $(A, B) \in \mathcal{P}(\mathcal{D}) \times \mathcal{P}(\mathcal{D})$,    $f(A \cup B) + f(A \cap B) \leq f(A) + f(B)$ .*

*Besides, $f$ is said to be* monotone *if for all $A \subseteq B \subseteq \mathcal{D}$, $f(A) \leq f(B)$. Otherwise, we say that $f$ is* non monotone.

### 2.2. Double-Greedy for USM

In this section, we outline Double-Greedy Algorithm (`DG`, Algorithm 1) from Buchbinder et al. (2012).

When maximizing a nonmonotone submodular function $f$, `DG` works in $d$ steps (one per item) and considers the items sequentially, th order being chosen by the user beforehand.

It first initializes a pair of sets $X_0 = \emptyset$ and $Y_0 = \mathcal{D}$ respectively as the empty set and the full set, and then modifies them sequentially.

At each step $i \in [d]$, `DG` looks at the "marginal gains" $\alpha_i$ and $\beta_i$ respectively corresponding to adding item $i$ to $X_{i-1}$ or removing it from $Y_{i-1}$ [Line 4-5]. It makes the decision of either adding or removing the item by sampling a Bernoulli random variable $K_i$ with parameter $p_i$, defined from the positive part of $\alpha_i$ and $\beta_i$ [Line 6-7]. After the $d$-th and last step, `DG` returns the set $X_d$, which is identical to $Y_d$ by construction [Line 14].

---

**Algorithm 1** Double-Greedy (`DG` from Buchbinder et al., 2012)

---

1: **Inputs:** $\mathcal{D}$ .
2: $(X_0, Y_0) \leftarrow (\emptyset, \mathcal{D})$ .
3: **for** $i = 1, \dots d$ **do**
4:     $\alpha_i \leftarrow f(X_{i-1} \cup \{i\}) - f(X_{i-1})$ .
5:     $\beta_i \leftarrow f(Y_{i-1} \setminus \{i\}) - f(Y_{i-1})$ .
6:     $p_i \leftarrow \frac{\max\{\alpha_i, 0\}}{\max\{\alpha_i, 0\} + \max\{\beta_i, 0\}}$ .
7:     $K_i \sim \mathcal{B}(p_i)$ .
8:     **if** $K_i$ **then**
9:         $(X_i, Y_i) \leftarrow (X_{i-1} \cup \{i\}, Y_{i-1})$ .
10:     **else**
11:         $(X_i, Y_i) \leftarrow (X_{i-1}, Y_{i-1} \setminus \{i\})$ .
12:     **end if**
13: **end for**
14: **Return:** $X_d \subseteq \mathcal{D}$ .

---

Overall, `DG` requires $4d$ calls to $f$ and satisfies the following guarantee.

**Theorem 1 (Buchbinder et al., 2012, Theorem I.2.)**   *Let $\mathcal{D}$ be a finite set. Algorithm `DG` returns a set $S$ such that*

$$\mathbb{E}\big[f(S)\big] \geq \frac{1}{2} f(A^*) .$$

The result being in expectation, one can repeatedly run `DG` to obtain an acceptable set with a high enough probability. In particular, we prove the following proposition in Appendix C.

**Proposition 1** *Let $\mathcal{D}$ be a finite set, $\delta > 0$ and $T \in \mathbb{N}^*$ such that $T > 2\log(1/\delta)$. If $(S_i)_{i \in [T]}$ is the sequence of sets obtained by running independently $T$ times Algorithm* DG, *then*

$$\max_{i \in [T]} f(S_i) > \left(\frac{1}{2} - \frac{\log(1/\delta)}{T}\right) f(A^*), \qquad w.p. \quad 1 - \delta.$$

**Stochastic bandit setting.** In our setting, using DG directly is not possible as we do not have access to the marginal gains $\alpha_i$ and $\beta_i$ but only to noisy estimates. To overcome this difficulty, Fourati et al. (2023) propose the *Randomized Greedy Learning* (RGL) algorithm, an *Explore-then-Commit* strategy satisfying a $O(dT^{2/3}\log(T)^{1/3})$ expected regret upper bound. Similarly to DG, RGL works in $d$ steps, one per item, each lasting $T^{2/3}\log(T)^{1/3}$ rounds. During the $i$-th step, RGL estimates the coefficients $\alpha_i$ and $\beta_i$, chooses a set $X_i$ (and $Y_i$) and move on to the next item. After $dT^{2/3}\log(T)^{1/3}$ exploration rounds, RGL commits to the last chosen set $X_d$.

However, we argue that RGL explores too much, and that logarithmic, problem-dependent regret upper bounds can be obtained both in expectation and with high-probability,

## 3. Full-bandit feedback algorithm: *Double-Greedy - Explore-then-Commit* (**DG−ETC**)

In this section, we propose *Double-Greedy - Explore-then-Commit* (DG-ETC), a novel algorithm for unconstrained submodular maximization (USM) with stochastic full bandit feedback. DG-ETC builds on insights from Buchbinder et al. (2012), Roughgarden and Wang (2018) and Harvey et al. (2020). We present the theoretical guarantees of DG-ETC in Section 4, which outperform existing approaches for this setting.

In the following, the word *round* refers to a single increment of time $t$, the word *step* refers to the per-item exploration steps (containing several rounds) and the word *phase* refers to the exploration/exploitation phases (the exploration phase containing one step per item).

### 3.1. Algorithms presentation

DG−ETC is presented in Algorithm 2, and is built on two subroutines: DG−Sp (Algorithm 3) to sample sets, and UpdExp (Algorithm 4) to update exploration parameters.

***Double-Greedy - Explore-then-Commit* (DG−ETC, Algorithm 2).** Algorithm DG−ETC is an algorithm implementing an *Explore-then-Commit* type strategy. It takes as inputs the set of items $\mathcal{D}$, the range of $f$ $c > 0$, the sub-Gaussian parameter of the noise $\sigma > 0$, as well as the horizon $T \in \mathbb{N}^*$ and a confidence level $\delta \in (0, 1)$. It first performs $d$ exploration steps (one per item in $\mathcal{D}$) [Lines 12 to 26], each lasting at most $4\tau_{\max}$ rounds where

$$\tau_{\max} = T^{2/3}\log(dT)^{1/3}. \tag{3}$$

Contrarily to RGL (Fourati et al., 2023), the duration of each exploration step is problem-adaptive, and can be considerably smaller than the aforementioned worst case (See Section 5.3). It then spends the rest of the rounds (at least $T^{1/3}\log(dT)^{2/3}$ ones) exploiting the collected information [Lines 27 to 32]. During this phase, it does not play a fixed set, but repeatedly samples random sets based on $d$ Bernoulli random variables with parameters $(p_j)_{j \in [d]}$ determined during the exploration phase.

---

**Algorithm 2** Double-Greedy - Explore-then-Commit (DG-ETC)

---

1: **Inputs:** $\mathcal{D}$, $c > 0$, $\sigma > 0$, $\delta > 0$, $T \in \mathbb{N}^*$.
2: /* Instantiating */
3: $d \leftarrow |\mathcal{D}|$.
4: Instantiate $g_{T,\delta}$ and $\tau_{\max}$ with (4) and (3).
5: Instantiate UpdExp with $g_{T,\delta}$ and $\tau_{\max}$.
6: /* Initialisation */
7: $(t, i) \leftarrow (1, 1)$.
8: $(\hat{\alpha}_j, \hat{\beta}_j)_{j \in [d]} \leftarrow 0$.
9: $(p_j)_{j \in [d]} \leftarrow 1/2$.
10: $(\tau_j)_{j \in [d]} \leftarrow 0$.
11: /* Exploration phase */
12: **while** $i \leq d$ **do**
13:   /* 4 rounds exploration */
14:   $(X_{i-1}, Y_{i-1}) \leftarrow$ DG-Sp$(\mathcal{D}, (p_j)_j, i)$.
15:   Play:
16:     $A_t \leftarrow X_{i-1}$, $A_{t+1} \leftarrow X_{i-1} \cup \{i\}$,
17:     $A_{t+2} \leftarrow Y_{i-1}$, $A_{t+3} \leftarrow Y_{i-1} \setminus \{i\}$.
18:   Receive:
19:     $Z_t$, $Z_{t+1}$, $Z_{t+1}$, $Z_{t+3}$.
20:   Update:
21:     $\hat{\alpha}_i \leftarrow \big(\tau_i \hat{\alpha}_i + (Z_{t+1} - Z_t)\big)/(\tau_i + 1)$,
22:     $\hat{\beta}_i \leftarrow \big(\tau_i \hat{\beta}_i + (Z_{t+3} - Z_{t+2})\big)/(\tau_i + 1)$,
23:     $\tau_i \leftarrow \tau_i + 1$,
24:     $(p_i, i) \leftarrow$ UpdExp$\big(i, (\hat{\alpha}_i, \hat{\beta}_i), \tau_i\big)$,
25:     $t \leftarrow t + 4$.
26: **end while**
27: /* Exploitation phase */
28: **while** $t \leq T$ **do**
29:   $(X_d, Y_d) \leftarrow$ DG-Sp$\big(\mathcal{D}, (p_j)_j, i\big)$.
30:   Play: $A_t \leftarrow X_d$.
31:   Update: $t \leftarrow t + 1$.
32: **end while**

---

***Double-Greedy - Sampling* (DG-Sp, Algorithm 3).** Both exploration and exploitation phases rely on the DG-Sp subroutine [Lines 14 and 29 in Algorithm 2], which is a variation of DG from Buchbinder et al. (2012) (Algorithm 1). DG-Sp relies on the parameters $(p_j)_{j \in [d]}$ provided by the meta-algorithm DG-ETC, which also provides a step $i \in \{1, \ldots, d, d+1\}$ before which DG-Sp should stop. Like DG, it begins by initializing two sets $X_0$ and $Y_0$ as the empty and the full sets. Then it iterates over the parameters $(p_j)_{j \in [d]}$ and pro-

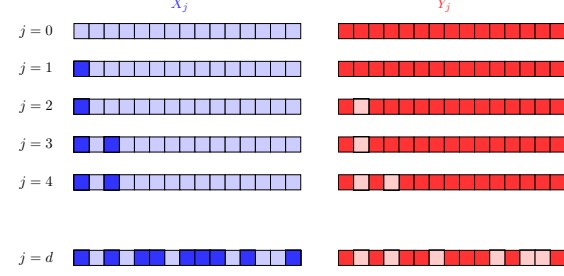

Figure 1: Example of sampling from DG-Sp, for $i = d + 1$ and $(K_j)_{j \in [d]} = (1, 0, 1, 0, \ldots 1)$.

ceeds to either add (to $X_{j-1}$) or remove (from $Y_{j-1}$) item $j$ in order to create $(X_j, Y_j)_{j < i}$, by sampling Bernoulli random variables. At the end, DG-Sp returns $(X_{i-1}, Y_{i-1})$ and DG-ETC then decides to either collect information when $i \leq d$ or exploit when $i = d+1$. An example of sampling from DG-Sp is illustrated in Figure 1.

**Exploration update for DG-ETC (UpdExp, Algorithm 4).** During the exploration, DG-ETC makes calls to Subroutine UpdExp [Line 24 in Algorithm 2]. The latter takes as inputs the index of the current step $i$, estimates of the marginal gains $(\alpha, \beta)$ and the current values of $\tau$ for item $i$. The objective of UpdExp is to check if we can determine an adequate Bernoulli parameter $p$ for item $i$ and/or if the exploration has lasted too long (if $\tau \geq \tau_{\max}$). In both those cases, UpdExp returns an adequate parameter $p$ and index $i + 1$ to tell DG-ETC to switch to the next item. Otherwise, $p$ stays the default $1/2$ and UpdExp returns current index $i$.

**Algorithm 3** Double-Greedy - Sampling (`DG-Sp`)

1: **Inputs:** $\mathcal{D}$, $(p_j) \in [0,1]^d$, $i \in [d+1]$.
2: $(X_0, Y_0) \leftarrow (\emptyset, \mathcal{D})$.
3: **for** $j = 1, \ldots, (i-1)$ **do**
4:     $K_j \sim \mathcal{B}(p_j)$.
5:     **if** $K_j$ **then**
6:         $(X_j, Y_j) \leftarrow (X_{j-1} \cup \{j\}, Y_{j-1})$.
7:     **else**
8:         $(X_j, Y_j) \leftarrow (X_{j-1}, Y_{j-1} \setminus \{j\})$.
9:     **end if**
10: **end for**
11: **Return:** $(X_{i-1}, Y_{i-1})$.

**Algorithm 4** Exploration update (`UpdExp`)

1: **Inputs:** $i \in [d]$, $(\alpha, \beta) \in [-c, c]^2$, $\tau \in \mathbb{N}^*$.
2: $\Lambda \leftarrow \{x \in [0,1] \text{ s.t. } \ell(\alpha, \beta, x) + \frac{g_{T,\delta}}{\sqrt{\tau}} \leq 0\}$.
3: $p \leftarrow 1/2$.
4: **if** $\Lambda \neq \emptyset$ **then**
5:     $p \leftarrow \arg\min_{x \in \Lambda} \ell(\alpha, \beta, x)$.
6:     $i \leftarrow i + 1$.
7: **else**
8:     **if** $\tau \geq \tau_{\max}$ **then**
9:         $p \leftarrow \frac{\alpha_+}{\alpha_+ + \beta_+}$ where $(\cdot)_+ = \max\{\cdot, 0\}$.
10:         $i \leftarrow i + 1$.
11:     **end if**
12: **end if**
13: **Return:** $(p, i)$.

### 3.2. Exploring just enough for zero exploitation regret: the key idea

In `DG-ETC`, the number of rounds devoted to the exploration for each item is adaptive, and is controlled by Subroutine `UpdExp`. Given estimated marginal gains $(\hat{\alpha}_i, \hat{\beta}_i)$ and an exploration time $\tau$, `UpdExp` checks if it is possible to counterbalance the (high-probability) errors coming from the different sources of uncertainties.

On the one hand, the per-round exploitation regret induced by all sources of uncertainty (estimations errors, random sampling, noise fluctuations) for item $i$, is bounded with high-probability (Proposition 2 in our analysis) by $\frac{g_{T,\delta}}{\sqrt{\tau_i}}$ where

$$g_{T,\delta} = \sqrt{2(2\sigma^2 + c^2)}\sqrt{2\log(dT) + \log(1/\delta)}\left(1 + 2\sqrt{\frac{\log(dT)}{T}} + \frac{9c}{\sqrt{2\sigma^2+c^2}}\left(\frac{\log(dT)}{T}\right)^{1/3}\right). \quad (4)$$

On the other hand, the decision to either add or remove item $i$ with probability $p_i$ [Line 30 in `DG-ETC` (Alg. 2) and Lines 4-9 in `DG-Sp` (Alg. 3)] induces an average loss[1] per exploration round bounded by $\ell(\hat{\alpha}_i, \hat{\beta}_i, p_i)$ where

$$\ell(\alpha, \beta, p) = \max\left(\ell^+(\alpha, \beta, p), \ell^-(\alpha, \beta, p)\right), \quad (5)$$

with $\ell^+(\alpha, \beta, p) = (1-p)\alpha - \frac{1}{2}(p\alpha + (1-p)\beta), \quad \ell^-(\alpha, \beta, p) = p\beta - \frac{1}{2}(p\alpha + (1-p)\beta)$.

In this definition, $\ell^+$ and $\ell^-$ are per-round regrets of using parameter $p$ when the (estimated) marginal gains are $(\alpha, \beta)$, corresponding to the two cases $\{i \in A^*\}$ and $\{i \notin A^*\}$. As one wants to hedge against both eventualities, we consider the worst-case loss $\ell$ which explains the max of both $\ell^+$ and $\ell^-$ in in Eq. (11).

`UpdExp` checks if, given estimations $(\hat{\alpha}_i, \hat{\beta}_i)$ and a current number of exploration rounds $4\tau_i$, it is possible to find a parameter $p_i$ so that the errors from uncertainties $\frac{g_{T,\delta}}{\sqrt{\tau_i}}$ are absorbed by the (hopefully negative) loss $\ell(\hat{\alpha}_i, \hat{\beta}_i, p_i)$. Formally, it looks for the existence of a $p_i \in [0, 1]$ so that

$$l(\hat{\alpha}_i, \hat{\beta}_i, p_i) + \frac{g_{T,\delta}}{\sqrt{\tau_i}} \leq 0, \quad (6)$$

---

1. Average with respect to the sampling variable $K_i$.

which is guaranteed to happen after a logarithmic number of rounds (Proposition 3). If it is the case, `UpdExp` returns this parameter $p_i$ and makes `DG-ETC` move on to the next item. Otherwise, the exploration for the current item $i$ continues unless it has already lasted too long (i.e. if $\tau_i \geq \tau_{\max}$). In this case, `UpdExp` returns parameter $p_i = \frac{\hat{\alpha}_{i,+}}{\hat{\alpha}_{i,+} + \hat{\beta}_{i,+}}$ and makes `DG-ETC` move on to the next step. This last choice for $p_i$ ensures the loss $\ell(\hat{\alpha}_i, \hat{\beta}_i, p_i)$ to be negative (or null) in the exploitation phase and the per-round regret for item $i$ to be bounded simply by $\frac{g_{T,\delta}}{\sqrt{\tau_{\max}}}$.

While `RGL` (Fourati et al., 2023) devotes the same number of rounds to all the items in the exploration phase, Subroutine `UpdExp` enables more flexibility. In particular, Section 4 links the number of exploration rounds necessary with problem-dependent quantities.

**Remark 1** *The possibility to counterbalance the accumulated errors with negative losses is enabled by the approximate regret criterion using the worst-case $1/2$ ratio, and an in-depth analysis of the original Double-Greedy algorithm. In all generality, this kind of intuition could also be applied to other methods to recover similar logarithmic upper bounds.*

## 4. Theoretical guarantees for `DG-ETC`

This section presents theoretical guarantees satisfied by our approach. We introduce a concept of problem-dependent *hardness* that characterizes how difficult it can be to maximize a given submodular function with our *Double-Greedy* approach. We then show that `DG-ETC` satisfies logarithmic $1/2$-approximate pseudo-regret upper bounds which depend on this hardness, with a $O(dT^{2/3} \log(dT)^{1/3})$ worst-case.

**Remark 2** *We remind that the items are taken in an arbitrary order, and the quantities may depend on it.*

### 4.1. Double-Greedy hardness

The following hardness notion relates to the sufficient number of exploration rounds that guarantees to find parameters $(p_i)_{i\in[d]}$ suitable to induce zero $1/2$-approximate regret during the exploitation.

**Definition 2 (DG-hardness)** *Let $\mathcal{D}$ be a set of $d$ elements (considered in an given order). Let $f$ be a submodular set-function over $\mathcal{D}$ and $i$ be an item in $\mathcal{D}$.*

*We define the* local DG-hardness *for item $i$ as*

$$h_{f,i} = \max_{X \subseteq [i-1]} \frac{\left(\alpha_f(i,X)_+ + \beta_f(i,X)_+\right)^2}{\left(\alpha_f(i,X)_+ - \beta_f(i,X)_+\right)^4},$$

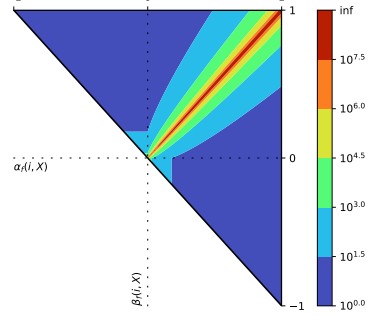

*where $(\,\cdot\,)_+ = \max\{\cdot, 0\}$ and*

$$\alpha_f(i,X) = f(X \cup \{i\}) - f(X),$$
$$\beta_f(i,X) = f\big((\mathcal{D} \setminus [i]) \cup X\big) - f\big((\mathcal{D} \setminus [i-1]) \cup X\big).$$

*We define the* global DG-hardness *as $H_f = \sum_{i\in[d]} h_{f,i}$.*

Figure 2: $h_{f,i}$ as a function of $\alpha_f(i,X)$ and $\beta_f(i,X)$ for $c = 1$.

**Remark 3**

- *This definition is actually not completely tight, as we will see in the analysis. But this form is more readable and convenient to use than the whole case disjunctions depending on the $(\alpha, \beta)$ configurations.*
- *We can also define a dual quantity, a* local DG-gap $\Delta_{f,i} = \left(h_{f,i}\right)^{-1/2}$, *playing the same role as the suboptimality gaps in pseudo-regret upper bounds for stochastic multi-armed bandits (it is homogeneous to a difference of rewards). The corresponding* global DG-gap *would be* $\Delta_f = H_f^{-1/2}$.

**Example** *For illustration purposes, let's consider the following function $g$ : we assume there exists $(\xi_i) \in [-1, 1]^d$ and $\nu \in (0, 1]$ such that for all $X \subseteq [d]$,*

$$g(X) = \left( \sum_{i \in X, \, \xi_i \geq 0} \xi_i \right)^{\nu} - \left( \sum_{i \in X, \, \xi_i < 0} -\xi_i \right)^{1/\nu} + \|\xi_-\|_1^{1/\nu}, \tag{7}$$

*where $\xi_- = (\xi_i \mathbb{1}\{\xi_i < 0\})_i$ and $\|\xi_-\|_1^{1/\nu}$ is here to guarantee the positivity of $g$. Then $g$ is submodular and for all $i \in [d]$:*

$$\Delta_{g,i} = \begin{cases} g([i]) - g([i-1]) & \text{if } \xi_i \geq 0, \\ g(\mathcal{D} \setminus [i]) - g(\mathcal{D} \setminus [i-1]) & \text{if } \xi_i < 0. \end{cases}$$

*These expressions remind the notion of suboptimality gaps common in bandit literature. If $\xi_i \geq 0$ then $i \in A^*$ and the DG-gap corresponds to the reward gained by adding $i$ to $[i-1]$. If $\xi_i < 0$ then $i \notin A^*$ and the DG-gap corresponds to the reward increase when removing $i$ from $\{i, i+1, \ldots, d\}$.*

*Notably, when $g$ is linear ($\nu = 1$), then the gaps $\Delta_{g,i} = \xi_i$ are independent from the ordering. They are intuitive as they correspond to the value of adding or removing the item, and the optimal set is the one containing all the items for which $\xi_i > 0$ (assuming that no item has a gap of $0$).*

### 4.2. Regret upper bounds for `DG-ETC`

This section presents our main result. We state a $1/2$-approximate pseudo-regret upper bounds for `DG-ETC`, the proof of which is outlined in Section 5.

**Theorem 2** *Let $\mathcal{D}$ be a finite set of $d \in \mathbb{N}^*$ items, $T \in \mathbb{N}^*$ a horizon with $d(T\sqrt{\log(dT)})^{2/3} \leq \frac{T}{2}$, $\sigma \in \mathbb{R}_+^*$ and $c \in \mathbb{R}_+^*$. Let $\delta > 0$.*

*Then, with probability greater than $1 - 10\delta/T$, `DG-ETC` satisfies*

$$R_T \leq C_1 \min \left\{ H_f \log(dT), \ dT^{2/3} \log(dT)^{1/3} \right\},$$

*where $C_1$ is a constant independent from $d$, $T$ and $\delta$.*

*Likewise, in expectation,*

$$\mathbb{E}[R_T] \leq C_2 \min \left\{ H_f \log(dT), \ dT^{2/3} \log(dT)^{1/3} \right\},$$

*where $C_1$ is a constant independent from $d$ and $T$.*

**Remark 4** *We can get more fine-grained bounds by using the local DG-hardnesses instead of the global one. From Eq.8 at the end of Section 5, we can keep the per-item granularity to get with probability at least $1 - 10\delta/T$*

$$R_T \leq C_3 \sum_{i \in [d]} \min \left\{ h_{f,i} \log(dT), \ T^{2/3} \log(dT)^{1/3} \right\},$$

*where $C_3$ is a constant independent from d, T and $\delta$.*

*In particular, depending on the scale of the horizon $T$ with respect to the different local hardnesses $(h_{f,i})_{i \in [d]}$, we obtain a mixed sum of some logarithmic terms, and others of magnitude $T^{2/3} \log(dT)^{1/3}$.*

## 5. Analysis of `DG-ETC`

This section presents a sketch of proof for Theorem 2.

We denote $\tau = \sum_{i \in [d]} 4\tau_i$ the last exploration round. For all the items $i$, we also denote $t_i = \sum_{j \leq i} 4\tau_i$, the last exploration round for item $i$. In this section, we have $i$-indices to denote items, and $t$-indices to denote that we place ourselves at round $t \in \mathbb{N}^*$. When $t$ is not made explicit (notably for $\hat{\alpha}_i$, $\hat{\beta}_i$ and $p_i$), it means that we place ourselves after round $t_i$ (and that these parameters are fixed).

**Outline of the proof.** The idea of the proof is to find a high-probability event (namely, $\mathcal{E}$) under which the exploration phase takes a logarithmic number of rounds per-item, and the regret is non positive during the exploitation phase. To that end, we first break the per-round regret of the exploitation phase down into per-item contributions (Section 5.1). Using this decomposition, we highlight an event $\mathcal{E}$ under which the per-round, per-item, regret is bounded by $l(\hat{\alpha}_i, \hat{\beta}_i, p_i) + \frac{g_{T,\delta}}{\sqrt{\tau_i}}$ for all the items $i$ (Section 5.2). Lastly, we prove that under $\mathcal{E}$, depending on the *DG-hardness* of $f$ (Definition 2), a logarithmic number of exploration rounds is sufficient to find a weights $p_i$ so that $l(\hat{\alpha}_i, \hat{\beta}_i, p_i) + \frac{g_{T,\delta}}{\sqrt{\tau_i}} \leq 0$ for all items $i$. Additionally, Subroutine `UpdExp` (Algorithm 4) returns a parameters $p_i$ so that $l(\hat{\alpha}_i, \hat{\beta}_i, p_i) \leq 0$ when $\tau_i$ reaches $\tau_{\max}$ for item $i$ (Section 5.3), so that the regret for this item remains bounded by $T^{2/3} \log(dT)^{1/3}$ when the estimation needs more rounds than what can be afforded.

**Template bound.** Let $\mathcal{E}$ be an event, defined later in Section 5.2. Then, the $1/2$-approximate pseudo-regret can be bounded as

$$R_T \leq \mathbb{1}_{\{\mathcal{E}^c\}} \frac{cT}{2} + \mathbb{1}_{\{\mathcal{E}\}} \left( 2c \sum_{i=1}^{d} \tau_i + \frac{1}{2} \sum_{t=\tau+1}^{T} r_t \right). \tag{8}$$

where $r_t = f(A^*) - 2f(A_t)$.

Under event $\mathcal{E}^c$, the pseudo-regret is upper bounded by a worst case $cT/2$. Under $\mathcal{E}$, each item $i$ uses $4\tau_i$ exploration rounds, each being bounded by a worst case $c/2$ regret, the rest of the rounds (between $\tau + 1$ and $T$) are devoted to the exploitation and their regret is upper bounded in the following. In particular, they yield no regret when the exploration is successful and the instance is "easy" enough.

### 5.1. Double-Greedy breakdown: Per-item exploitation regrets

We use an approach similar to Buchbinder et al. (2012) to bound the per-round exploitation regret $r_t$ with a sum of per-item contributions.

**Item-wise breakdown.** Let $t > \tau$. We considers sets $(A_{i,t}^*)_{i \in [d]}$, with $A_{0,t}^* = A^*$ and $A_{d,t}^* = A_t$, constructed to control the evolution of $(f(A_{i,t}))_{i \in [d]}$ from $f(A^*)$ to $f(A_t)$ using the coefficients $(\alpha_{i,t}, \beta_{i,t})_{i \in [d]}$. We define

$$
\begin{aligned}
&\text{For } i = 0, & A_{0,t}^* &= A^*, & &\text{with } X_{0,t} = \emptyset, \quad Y_{0,t} = \mathcal{D}, \\
&\forall i \in [d], & A_{i,t}^* &= (A^* \cup X_{i,t}) \cap Y_{i,t}, & &\text{with } X_{i,t} \subseteq A_{i,t}^* \subseteq Y_{i,t}, \\
&\text{For } i = d, & A_{d,t}^* &= X_{d,t} = Y_{d,t} = A_t,
\end{aligned}
\tag{9}
$$

where $X_{i,t} = \{j \leq i, K_{j,t} = 1\}$ and $Y_{i,t} = \mathcal{D} \setminus \{j \leq i, K_{j,t} = 0\}$ are the sets defined in Subroutine `DG-Sp` (Algorithm 3).

Using these sets and the definition of $r_t$ in Eq. (8), a telescopic argument yields

$$
\begin{aligned}
r_t &\leq f(A^*) - f(A_t) - \frac{1}{2}\Big[ 2f(A_t) - (f(\emptyset) + f(\mathcal{D})) \Big] & &\leftarrow (f \geq 0) \\
&= \Big[ f(A_{0,t}^*) - f(A_{d,t}^*) \Big] - \frac{1}{2}\Big[ f(X_{d,t}) - f(X_{0,t}) + f(Y_{d,t}) - f(Y_{0,t}) \Big] \\
&= \sum_{i=1}^d \Big[ f(A_{i-1,t}^*) - f(A_{i,t}^*) - \frac{1}{2}\big( K_{i,t}\alpha_{i,t} + (1 - K_{i,t})\beta_{i,t} \big) \Big],
\end{aligned}
\tag{10}
$$

where for all $i \in [d]$, $\alpha_{i,t} = f(X_{i-1,t} \cup \{i\}) - f(X_{i-1,t})$,
$$\beta_{i,t} = f(Y_{i-1,t} \setminus \{i\}) - f(Y_{i-1,t}).$$

**Submodularity.** While the marginal gains $(\alpha_{i,t}, \beta_{i,t})_{i \in [d]}$ can be estimated, the sets $A^*$, and $(A_{i,t}^*)_{i \in [d]}$ remain unknown. However, the definition of $(A_{i,t}^*)_{i \in [d]}$ and submodularity yield
- If $\{i \in A^*\}$, then $f(A_{i-1,t}^*) - f(A_{i,t}^*) \leq (1 - K_{i,t})\alpha_{i,t}$,
- Else $\{i \notin A^*\}$, and $f(A_{i-1,t}^*) - f(A_{i,t}^*) \leq K_{i,t}\beta_{i,t}$.

Using these inequalities, Eq. (10) becomes

$$
r_t \leq \sum_{i \in [d]} \Big[ \mathbb{1}_{\{i \in A^*\}}(1 - K_{i,t})\alpha_{i,t} + \mathbb{1}_{\{i \notin A^*\}}K_{i,t}\beta_{i,t} - \frac{1}{2}\big( K_{i,t}\alpha_{i,t} + (1 - K_{i,t})\beta_{i,t} \big) \Big].
$$

Since $\{i \in A^*\}$ and $\{i \notin A^*\}$ are exclusive events, we have

$$
\sum_{t=\tau+1}^T r_t \leq \sum_{i \in [d]} \max\big\{ R_{T,i}^+, R_{T,i}^- \big\},
\tag{11}
$$

where $R_{T,i}^+ = \sum_{t=\tau+1}^T \Big[ (1 - K_{i,t})\alpha_{i,t} - \frac{1}{2}\big( K_{i,t}\alpha_{i,t} + (1 - K_{i,t})\beta_{i,t} \big) \Big]$,
$R_{T,i}^- = \sum_{t=\tau+1}^T \Big[ K_{i,t}\beta_{i,t} - \frac{1}{2}\big( K_{i,t}\alpha_{i,t} + (1 - K_{i,t})\beta_{i,t} \big) \Big]$.

## 5.2. High-probability exploitation regret

Let $i \in [d]$, the objective now is to control $\max\left\{R_{T,i}^+,\ R_{T,i}^-\right\}$ from Eq. (11). To that end, the following proposition (proven in Appendix D.1) states how the errors coming from the different randomness sources concentrate.

**Proposition 2** *Let $\mathcal{H}$ and $\mathcal{E}$ be the event*

$$\mathcal{H} = \begin{cases} \forall i \in [d]\,, \ \forall t > t_{i-1}\,, & |\bar{\alpha}_i - \hat{\alpha}_{i,t}| \leq \sqrt{2\sigma^2 + c^2}\sqrt{2\frac{\log(dT/\delta)+\log(1+4\tau_{i,t})}{\tau_{i,t}+1}}\,, \\ & |\bar{\beta}_i - \hat{\beta}_{i,t}| \leq \sqrt{2\sigma^2 + c^2}\sqrt{2\frac{\log(dT/\delta)+\log(1+4\tau_{i,t})}{\tau_{i,t}+1}} \end{cases},$$

$$\mathcal{E} = \mathcal{H} \cap \left\{ \forall i \in [d]\,, \quad \max\left\{R_{T,i}^+, R_{T,i}^-\right\} - (T-\tau)\left(l(\hat{\alpha}_i,\ \hat{\beta}_i,\ p_i) + \frac{g_{T,\delta}}{\sqrt{\tau_i}}\right) \leq 0 \right\},$$

*where for all $i \in [d]$, $\bar{\alpha}_i = \mathbb{E}\left[\alpha_{i,t}|(p_j)_{j<i}\right]$ and $\bar{\beta}_i = \mathbb{E}\left[\beta_{i,t}|(p_j)_{j<i}\right]$, both quantities being constant for rounds $t > t_{i-1}$, and $g_{T,\delta}$ is defined in Eq. (4).*
   *Then, $\mathbb{P}(\mathcal{H}^c) \leq \frac{4\delta}{T}$, and $\mathbb{P}(\mathcal{E}^c) \leq \frac{10\delta}{T}$.*

**Template bound.** Reinjecting Eq. (11) and Proposition 2 yields

$$R_T \leq \mathbb{1}_{\{\mathcal{E}^c\}}\frac{cT}{2} + \mathbb{1}_{\{\mathcal{E}\}}\sum_{i\in[d]}\left(2c\tau_i + (T-\tau)\left(l(\hat{\alpha}_i,\hat{\beta}_i,p_i) + \frac{g_{T,\delta}}{\sqrt{\tau_i}}\right)\right), \qquad (12)$$

where $\mathcal{E}$ is the event defined Proposition 2.

## 5.3. Sufficient exploration

In this section, we analyze the exploration steps for each item and we exhibit sufficient conditions for them to only last a logarithmic number of rounds. The default choice of $p_i = \frac{\alpha_{i,+}}{\alpha_{i,+}+\beta_{i,+}}$ when $\tau_i \geq \tau_{\max}$ in Subroutine $\mathtt{UpdExp}$ (Algorithm 4) ensures $\ell(\hat{\alpha}_i,\ \hat{\beta}_i,\ p_i) \leq 0$, which in turns yield a $O\left(T\sqrt{\log(dT)}^{2/3}\right)$ regret upper bound for item $i$.

   Subroutine $\mathtt{UpdExp}$ looks for a parameter $p \in [0,1]$ so that both

$$\left(1-p\right)\hat{\alpha}_i - \frac{1}{2}\left(p\hat{\alpha}_i + (1-p)\hat{\beta}_i\right) \leq -\frac{g_{T,\delta}}{\sqrt{\tau_i}}\,, \quad \text{and} \quad p\hat{\beta}_i - \frac{1}{2}(p\hat{\alpha}_i + (1-p)\hat{\beta}_i) \leq -\frac{g_{T,\delta}}{\sqrt{\tau_i}}\,. \quad (13)$$

Under $\mathcal{E}$, as we can upper bound $|\hat{\alpha}_i - \bar{\alpha}_i|$ and $|\hat{\beta}_i - \bar{\beta}_i|$ (Proposition 2), it is sufficient to have

$$\begin{cases} \left(1-p\right)\bar{\alpha}_i - \frac{1}{2}(p\bar{\alpha}_i + (1-p)\bar{\beta}_i) & \leq -\frac{g_{T,\delta}}{\sqrt{\tau_i}} - \frac{3}{2}\sqrt{2\sigma^2 + c^2}\sqrt{2\frac{\log(dT/\delta)+\log(1+4\tau_i)}{\tau_i+1}} \\ p\bar{\beta}_i - \frac{1}{2}(p\bar{\alpha}_i + (1-p)\bar{\beta}_i) & \leq -\frac{g_{T,\delta}}{\sqrt{\tau_i}} - \frac{3}{2}\sqrt{2\sigma^2 + c^2}\sqrt{2\frac{\log(dT/\delta)+\log(1+4\tau_i)}{\tau_i+1}} \end{cases},$$

for which it is in turn sufficient to have

$$p(\bar{\beta}_i - 3\bar{\alpha}_i) \leq -\frac{g_i + \gamma_{T,\delta}}{\sqrt{\tau_i}} + (\beta_i - 2\bar{\alpha}_i), \quad \text{and} \quad p(3\bar{\beta}_i - \bar{\alpha}_i) \leq -\frac{g_i + \gamma_{T,\delta}}{\sqrt{\tau_i}} + \bar{\beta}_i\,, \qquad (14)$$

where $\gamma_{T,\delta} = 3\sqrt{(2\sigma^2 + c^2)(\log(dT/\delta) + \log(1+T))}$.
   The following proposition gives sufficient conditions to find a $p_i$ for Eq. (13) to be satisfied.

**Proposition 3** *For each items $i \in [d]$, under event $\mathcal{E}$ defined in Proposition 2, $\mathtt{UpdExp}$ finds a weight $p_i$ such that $l(\hat{\alpha}_{i,t},\ \hat{\beta}_{i,t},\ p_i) + \frac{g_{T,\delta}}{\sqrt{\tau_{i,t}}} \leq 0$ before $4\tau_{i,t}$ the current number of exploration rounds for item $i$ has reached the value $4(g_{T,\delta} + \gamma_{T,\delta})^2\,h_{f,i}$.*

**Template bound.** Using Proposition 3, the upper bound Eq. (12) becomes

$$R_T \leq \mathbb{1}_{\{\mathcal{E}^c\}} \frac{cT}{2} + \mathbb{1}_{\{\mathcal{E}\}} \sum_{i \in [d]} \left[ 2c \min \left\{ (g_{T,\delta} + \gamma_{T,\delta})^2 h_{f,i} , \ \tau_{\max} \right\} \right.$$

$$\left. + \mathbb{1}_{\left\{ (g_{T,\delta} + \gamma_{T,\delta})^2 h_{f,i} > \tau_{\max} \right\}} \tau_{\max} \frac{T g_{T,\delta}}{(\tau_{\max})^{3/2}} \right]$$

$$= \mathbb{1}_{\{\mathcal{E}^c\}} \frac{cT}{2} + \mathbb{1}_{\{\mathcal{E}\}} \sum_{i \in [d]} \left( 2c + \frac{g_{T,\delta}}{\log(dT)^{1/2}} \right) \min \left\{ (g_{T,\delta} + \gamma_{T,\delta})^2 h_{f,i} , \ \tau_{\max} \right\}, \quad (15)$$

where $\tau_{\max} = T^{2/3} \log(dT)^{1/3}$ is defined in Eq.(3).

The high-probability result comes from event $\mathcal{E}$ happening with probability greater than $1 - \frac{10\delta}{T}$ (Proposition 2). Choosing $\delta = 1$ yields the bound in expectation.

## 6. Concluding remarks

We propose and analyze Algorithm DG-ETC (Algorithm 2) for the online unconstrained submodular maximization problem, with stochastic bandit feedback. Our algorithm is a considerable improvement from other existing approaches, as it satisfies logarithmic upper bounds for the $1/2$-approximate pseudo-regret, dependent on a new notion of hardness that we introduce. Possible extensions include designing anytime variants, and algorithms adaptive to the adversarial/stochastic setting (best-of-both worlds).

An interesting feature of DG-ETC is that it leverages the looseness of worst-case approximation ratios in non-adversarial cases, and we argue that this kind of strategy could also be applied to other settings to yields similar performances.

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

## Appendix A. Extended related works

This section completes the related work of the main paper in Section 1.3.

We discuss additional submodular optimization settings existing in the literature, namely, offline and online minimization, constrained maximization, as well as other maximization frameworks.

**Submodular optimization (offline).** Submodularity is also studied in settings different from the unconstrained non monotone maximization case that we look at.

*Minimization* can be solved in polynomial time (Grötschel et al., 1981) and there is an extensive line of work studying that setting. See Lee et al. (2015); Chakrabarty et al. (2017); Axelrod et al. (2020); Jiang (2021, 2022); Jiang et al. (2023) for the most recent ones.

For *maximization*, the constrained non monotone setting is even more challenging and there is a line of work constantly improving the approximations (Lee et al., 2009; Chekuri et al., 2011; Buchbinder et al., 2014; Vondrák, 2013; Ene and Nguyen, 2016; Buchbinder and Feldman, 2019; Tukan et al., 2024; Buchbinder and Feldman, 2024), which is known to be smaller than $.478$ in polynomial time (Qi, 2024). In the monotone setting, there also exist a line of works interested in the identification of the best subset with the minimal number of potentially noisy calls to the function (Singla et al., 2016; Hassidim and Singer, 2017; Karimi et al., 2017; Hassidim and Singer, 2018). Other papers also study maximization of a submodular function only known from given samples (Balkanski et al., 2016, 2017).

**Online and bandit submodular optimization.** Both maximization and minimization are explored in the online learning and bandit literature.

For the *minimization* version, Hazan and Kale (2012) introduces an online adversarial setting and proposes an algorithm with sublinear regret. Its results are improved in Matsuoka et al. (2021) and Ito (2022), the latter also proposing results for the bandit setting. In particular, a commonly used tool is the Lovász extension, reducing the problem to convex minimization.

The *maximization* version is studied in Streeter and Golovin (2008) with a resource allocation perspective, giving guarantees for the $(1 - 1/e)$-approximate expected regret in the monotone setting. Its results are extended to matroid constraints in Streeter et al. (2009); Golovin et al. (2014), and improved in Harvey et al. (2020) using curvatures. The case of bandit feedback is also studied for monotone functions. Yue and Guestrin (2011) and Guillory and Bilmes (2011) are early works works providing theoretical guarantees. They are followed by several papers considering variants of this setting (Kohli et al., 2013; Gabillon et al., 2013; Chen et al., 2018).

## Appendix B. Reminders on sub-Gaussianity

We use sub-Gaussianity assumptions and common concentration tools to control deviations of the noise $(\eta_t)_{t \in [T]}$ and the randomization process of our approach. This section remind useful results.

**Definition 3 (Sub-Gaussian)** *Let $\sigma > 0$ and $X$ be a real-valued random variable such that $\mathbb{E}[X] = 0$. We say that $X$ is $\sigma^2$-sub-Gaussian, for all $\lambda \in \mathbb{R}$,*

$$\mathbb{E}[\exp(\lambda X)] \leq \exp\left(\frac{\lambda^2 \sigma^2}{2}\right).$$

In particular, for bounded independent random variables, we have the following lemma.

**Lemma 1 (Hoeffding's inequality for sum of i.i.d. bounded r.v.)** *Let $\delta > 0$, $N \in \mathbb{N}^*$, and $(Z_n)_{n \in [N]}$ a family of i.i.d. real random variables bounded in $[a, b]$ where $(a, b) \in (\mathbb{R})^2$, with mean $\mu \in [a, b]$.*
*Then for all $n \in [N]$, $Z_n$ is $\frac{(b-a)^2}{4}$-sub-Gaussian, and with probability at least $1 - \delta$,*

$$\frac{1}{N} \sum_{n=1}^{N} \left[Z_n - \mu\right] < \frac{b-a}{2} \sqrt{\frac{2}{N} \log(1/\delta)}.$$

The sub-Gaussianity for bounded random variables an the concentration for the sums of i.i.d random variables are classical results proven that can be found Wainwright (2019) for example.

As we estimate quantities in an online setting, with observations arriving sequentially and depending on our actions, we need a more powerful tool. This is provided by the following lemma.

**Lemma 2 (Hoeffding's inequality with martingales)** *Let $\delta > 0$, $\sigma > 0$. Let $(\mathcal{G}_t)_{t \in \mathbb{N}}$ be a filtration and $(Z_t)_{t \in \mathbb{N}^*}$ a $(\mathcal{G}_t)$-adapted martingale with $\mathbb{E}[Z_1] = 0$. We assume that for all $t \in \mathbb{N}$, $Z_{t+1}$ is $\sigma^2$-sub-Gaussian conditionally to $\mathcal{G}_t$. Let $(U_t)_{t \in \mathbb{N}^*}$ be a $(\mathcal{G}_t)$-predictable process. Then, with probability at least $1 - \delta$, for all $t \in \mathbb{N}$*

$$\frac{\sum_{s=1}^{t} U_s Z_s}{1 + \sum_{s=1}^{t} U_s^2} < \frac{\sigma}{\sqrt{1 + \sum_{s=1}^{t} U_s^2}} \sqrt{2 \log(1/\delta) + \log\left(1 + \sum_{s=1}^{t} U_s^2\right)}$$

The proof relies on the method of mixture, widely used in the bandit literature (Abbasi-Yadkori et al., 2011; Faury et al., 2020; Zhou et al., 2024).
**Proof** Let $\delta > 0$, $\sigma > 0$. Let $(\mathcal{G}_t)$ be a filtration and $(Z_t)$ be a $\mathcal{G}_t$-adapted martingale with $\mathbb{E}[Z_1] = 0$ and so that for all $t \in \mathbb{N}$, $Z_{t+1}$ is $\sigma^2$-sub-Gaussian conditionally to $\mathcal{G}_t$. Let $(U_t)$ be a $\mathcal{G}_t$-predictable process.

Let $t \in \mathbb{N}^*$, a first direct result is that, $U_t Z_t$ is $(\sigma U_t)^2$-sub-Gaussian conditionally to $\mathcal{G}_{t-1}$. Let $\lambda \in \mathbb{R}$. Then,

$$\mathbb{E}\left[\exp\left(\lambda U_t Z_t - \frac{\lambda^2}{2}(\sigma U_t)^2\right) | \mathcal{G}_{t-1}\right] \leq 1. \tag{16}$$

We define

$$M_t(\lambda) = \exp\left(\lambda \sum_{s=1}^{t} U_s Z_s - \frac{\lambda^2}{2} \sum_{s=1}^{t} (\sigma U_s)^2\right)$$

with $M_0(\lambda) = 1$. From eq. (16),

$$\forall t \in \mathbb{N}, \quad \mathbb{E}[M_t(\lambda)|\mathcal{G}_t] = \mathbb{E}\left[\exp\left(\lambda \sum_{s=1}^t U_s Z_s - \frac{\lambda^2}{2}\sum_{s=1}^t(\sigma U_s)^2\right)\Big|\mathcal{G}_{t-1}\right]$$

$$= M_{t-1}(\lambda)\,\mathbb{E}\left[\exp\left(\lambda U_t Z_t - \frac{\lambda^2}{2}(\sigma U_t)^2\right)\Big|\mathcal{G}_{t-1}\right]$$

$$\leq M_{t-1}(\lambda)\,.$$

Then, $(M_t(\lambda))_t$ is a $\mathcal{G}_t$-supermartingale, with $\mathbb{E}[M_t(\lambda)] \leq 1$.

We now consider $\lambda \sim \mathcal{N}(0, 1/\sigma^2)$, independent from all the other distributions, then we can define

$$\bar{M}_t = \mathbb{E}_{\lambda \sim \mathcal{N}(0,1/\sigma^2)}[M_t(\lambda)]$$

$$= \frac{\sigma}{\sqrt{2\pi}}\int_{\mathbb{R}}\exp\left(-\frac{(\sigma x)^2}{2}\right)\exp\left(x\sum_{s=1}^t U_s Z_s - \frac{x^2}{2}\sum_{s=1}^t(\sigma U_s)^2\right)dx$$

$$= \frac{\sigma}{\sqrt{2\pi}}\int_{\mathbb{R}}\exp\left(-\frac{(\sigma x)^2(1+\sum_{s=1}^t U_s^2)}{2} + x\sum_{s=1}^t U_s Z_s\right)dx$$

$$= \frac{\sigma}{\sqrt{2\pi}}\int_{\mathbb{R}}\exp\left(-\frac{\sigma^2(1+\sum_{s=1}^t U_s^2)}{2}\left(x^2 - 2x\frac{\sum_{s=1}^t U_s Z_s}{\sigma^2(1+\sum_{s=1}^t U_s^2)}\right)\right)dx$$

$$= \frac{\sigma}{\sqrt{2\pi}}\int_{\mathbb{R}}\exp\left(-\frac{\sigma^2(1+\sum_{s=1}^t U_s^2)}{2}\left(x - \frac{\sum_{s=1}^t U_s Z_s}{\sigma^2(1+\sum_{s=1}^t U_s^2)}\right)^2 + \frac{(\sum_{s=1}^t U_s Z_s)^2}{2\sigma^2(1+\sum_{s=1}^t U_s^2)}\right)dx$$

$$= \exp\left(\frac{(\sum_{s=1}^t U_s Z_s)^2}{2\sigma^2(1+\sum_{s=1}^t U_s^2)}\right)\frac{\sigma}{\sqrt{2\pi}}\frac{\sqrt{2\pi}}{\sigma\sqrt{1+\sum_{s=1}^t U_s^2}}\frac{\sigma\sqrt{1+\sum_{s=1}^t U_s^2}}{\sqrt{2\pi}}$$

$$\int_{\mathbb{R}}\exp\left(-\frac{\sigma^2(1+\sum_{s=1}^t U_s^2)}{2}\left(x - \frac{\sum_{s=1}^t U_s Z_s}{\sigma^2(1+\sum_{s=1}^t U_s^2)}\right)^2\right)dx$$

$$= \exp\left(\frac{(\sum_{s=1}^t U_s Z_s)^2}{2\sigma^2(1+\sum_{s=1}^t U_s^2)}\right)\frac{1}{\sqrt{1+\sum_{s=1}^t U_s^2}}\underbrace{\mathbb{E}_{\lambda \sim \mathcal{N}\left(\frac{\sum_{s=1}^t U_s Z_s}{\sigma^2(1+\sum_{s=1}^t U_s^2)}, \frac{1}{\sigma^2(1+\sum_{s=1}^t U_t^2)}\right)}[1]}$$

$$= \frac{1}{\sqrt{1+\sum_{s=1}^t U_s^2}}\exp\left(\frac{(\sum_{s=1}^t U_s Z_s)^2}{2\sigma^2(1+\sum_{s=1}^t U_s^2)}\right)$$

$$\bar{M}_t = \exp\left(\frac{(\sum_{s=1}^t U_s Z_s)^2}{2\sigma^2(1+\sum_{s=1}^t U_s^2)} - \frac{1}{2}\log\left(1+\sum_{s=1}^t U_s^2\right)\right)$$

Besides,

$$
\begin{aligned}
\mathbb{E}\Big[\bar{M}_t\big|\mathcal{G}_{t-1}\Big] &= \mathbb{E}\Big[\mathbb{E}_{\lambda\sim\mathcal{N}(0,1/\sigma^2)}[M_t(\lambda)]\big|\mathcal{G}_{t-1}\Big] \\
&= \mathbb{E}_{\lambda\sim\mathcal{N}(0,1/\sigma^2)}\Big[\mathbb{E}[M_t(\lambda)|\mathcal{G}_{t-1}]\Big] \\
&\leq \mathbb{E}_{\lambda\sim\mathcal{N}(0,1/\sigma^2)}\Big[M_{t-1}(\lambda)\Big] \\
&= \bar{M}_{t-1} \, .
\end{aligned}
$$

So $(\bar{M}_t)_t$ is also a supermartingale, which yield that

$$
\mathbb{E}[\bar{M}_t] \leq \mathbb{E}[\bar{M}_0] = 1 \, .
$$

Let $u_t > 0$. Now, using Chernoff's method,

$$
\begin{aligned}
\mathbb{P}\Bigg(\frac{\sum_{s=1}^t U_s Z_s}{1 + \sum_{s=1}^t U_s^2} \geq u_t\Bigg) &\leq \mathbb{P}\Bigg(\exp\Big(\frac{(\sum_{s=1}^t U_s Z_s)^2}{2\sigma^2(1 + \sum_{s=1}^t U_s^2)} - \frac{u_t^2}{2\sigma^2}(1 + \sum_{s=1}^t U_s^2)\Big) \geq 1\Bigg) \\
&\leq \mathbb{E}\Bigg[\exp\Big(\frac{(\sum_{s=1}^t U_s Z_s)^2}{2\sigma^2(1 + \sum_{s=1}^t U_s^2)} - \frac{u_t^2}{2\sigma^2}(1 + \sum_{s=1}^t U_s^2)\Big)\Bigg] \\
&\leq \mathbb{E}\Bigg[\bar{M}_t \exp\Big(\frac{1}{2}\log(1 + \sum_{s=1}^t U_s^2) - \frac{u_t^2}{2\sigma^2}(1 + \sum_{s=1}^t U_s^2)\Big)\Bigg] \, .
\end{aligned}
$$

Choosing $u_t = \frac{\sigma}{\sqrt{1 + \sum_{s=1}^t U_s^2}}\sqrt{2\log(1/\delta) + \log(1 + \sum_{s=1}^t U_s^2)}$,

$$
\begin{aligned}
\mathbb{P}\Bigg(\frac{\sum_{s=1}^t U_s Z_s}{1 + \sum_{s=1}^t U_s^2} \geq u_t\Bigg) &\leq \mathbb{E}\Big[\delta\bar{M}_t\Big] \\
&\leq \delta \, .
\end{aligned}
$$

The bound for all $t$ is based on the stopping time construction from Abbasi-Yadkori et al. (2011). ■

## Appendix C. Proof for the high-probability bound of Double-Greedy (Algorithm 1, DG from Buchbinder et al. (2012))

**Proposition 1** *Let $\mathcal{D}$ be a finite set, $\delta > 0$ and $T \in \mathbb{N}^*$ such that $T > 2\log(1/\delta)$. If $(S_i)_{i \in [T]}$ is the sequence of sets obtained by running independently $T$ times Algorithm DG, then*

$$\max_{i \in [T]} f(S_i) > \left( \frac{1}{2} - \frac{\log(1/\delta)}{T} \right) f(A^*), \qquad w.p. \quad 1 - \delta.$$

**Proof** Let $1 > \delta > 0$ and $T \in \mathbb{N}^*$ such that $T > 2\log(1/\delta)$. Then $(f(S_i))_{i \in [T]}$ is a sequence of $T$ i.i.d. random variables, bounded in $[0, f(A^*)]$.

Let $\frac{1}{2} > u > 0$. Then

$$
\begin{aligned}
\mathbb{P}\left( \max_{i \in [T]} f(S_i) < \left( \frac{1}{2} - u \right) f(A^*) \right) &= \mathbb{P}\left( \forall i \in [T], \; f(S_i) < \left( \frac{1}{2} - u \right) f(A^*) \right) \\
&= \prod_{i=1}^{T} \mathbb{P}\left( f(S_i) < \left( \frac{1}{2} - u \right) f(A^*) \right) \\
&\leq \mathbb{P}\left( \left( \frac{1}{2} - u \right) f(A^*) + f(A^*) - f(S_1) > f(A^*) \right)^T \\
&\leq \frac{1}{f(A^*)^T} \mathbb{E}\left[ \left( \frac{1}{2} - u \right) f(A^*) + f(A^*) - f(S_1) \right]^T \quad \leftarrow \text{Markov} \\
&\leq \frac{1}{f(A^*)^T} \left[ \left( \frac{1}{2} - u \right) f(A^*) + \frac{1}{2} f(A^*) \right]^T \quad \leftarrow \text{Theorem 1} \\
&= (1 - u)^T \\
&\leq \exp(-Tu).
\end{aligned}
$$

Therefore, taking $u = \frac{\log(1/\delta)}{T}$, we have the result

$$\mathbb{P}\left( \max_{i \in [T]} f(S_i) < \left( \frac{1}{2} - u \right) f(A^*) \right) \leq \delta.$$

∎

## Appendix D. Proofs for the analysis of Double-Greedy - Explore-Then-Commit (`DG-ETC`, ours)

### D.1. Proof for the high-probability exploitation regret

**Proposition 2** *Let $\mathcal{H}$ and $\mathcal{E}$ be the event*

$$\mathcal{H} = \begin{cases} \forall i \in [d], \ \forall t > t_{i-1}, & |\bar{\alpha}_i - \hat{\alpha}_{i,t}| \leq \sqrt{2\sigma^2 + c^2}\sqrt{2\frac{\log(dT/\delta)+\log(1+4\tau_{i,t})}{\tau_{i,t}+1}}, \\ & |\bar{\beta}_i - \hat{\beta}_{i,t}| \leq \sqrt{2\sigma^2 + c^2}\sqrt{2\frac{\log(dT/\delta)+\log(1+4\tau_{i,t})}{\tau_{i,t}+1}} \end{cases},$$

$$\mathcal{E} = \mathcal{H} \cap \left\{ \forall i \in [d], \quad \max\left\{R_{T,i}^+, R_{T,i}^-\right\} - (T-\tau)\Big(l(\hat{\alpha}_i, \ \hat{\beta}_i, \ p_i) + \frac{g_{T,\delta}}{\sqrt{\tau_i}}\Big) \leq 0 \right\},$$

*where for all $i \in [d]$, $\bar{\alpha}_i = \mathbb{E}[\alpha_{i,t}|(p_j)_{j<i}]$ and $\bar{\beta}_i = \mathbb{E}[\beta_{i,t}|(p_j)_{j<i}]$, both quantities being constant for rounds $t > t_{i-1}$, and $g_{T,\delta}$ is defined in Eq. (4).*
*Then, $\mathbb{P}(\mathcal{H}^c) \leq \frac{4\delta}{T}$, and $\mathbb{P}(\mathcal{E}^c) \leq \frac{10\delta}{T}$.*

**Proof** We remind Eq. (11),

$$\sum_{t=\tau+1}^T r_t \leq \sum_{i\in[d]} \max\left\{R_{T,i}^+, \ R_{T,i}^-\right\}, \tag{11}$$

where $R_{T,i}^+ = \sum_{t=\tau+1}^T \left[(1-K_{i,t})\alpha_{i,t} - \frac{1}{2}\big(K_{i,t}\alpha_{i,t} + (1-K_{i,t})\beta_{i,t}\big)\right]$,

$R_{T,i}^- = \sum_{t=\tau+1}^T \left[K_{i,t}\beta_{i,t} - \frac{1}{2}\big(K_{i,t}\alpha_{i,t} + (1-K_{i,t})\beta_{i,t}\big)\right]$.

For $i \in [d]$, we define $\bar{\alpha}_i = \mathbb{E}[\alpha_{i,t}|(p_j)_{j<i}]$ and $\bar{\beta}_i = \mathbb{E}[\beta_{i,t}|(p_j)_{j<i}]$, both quantities being constant for rounds $t > t_{i-1}$ (and thus, for $t > \tau$). Separating the different sources of randomness yields

$$R_{T,i}^+ = \bar{E}_{T,i}^+ + \hat{E}_{T,i}^+ + L_{T,i}^+, \qquad R_{T,i}^- = \bar{E}_{T,i}^- + \hat{E}_{T,i}^- + L_{T,i}^-,$$

where we have

- errors coming from the deviation of $(\alpha_{i,t}, \ \beta_{i,t})$ from $(\bar{\alpha}_i, \ \bar{\beta}_i)$:
  $\bar{E}_{T,i}^+ = \sum_{t=\tau+1}^T \left[(1-K_{i,t})(\alpha_{i,t}-\bar{\alpha}_i) - \frac{1}{2}\big(K_{i,t}(\alpha_{i,t}-\bar{\alpha}_i) + K_{i,t}^c(\beta_{i,t}-\bar{\beta}_i)\big)\right]$,
  $\bar{E}_{T,i}^- = \sum_{t=\tau+1}^T \left[K_{i,t}(\beta_{i,t}-\bar{\beta}_i) - \frac{1}{2}\big(K_{i,t}(\alpha_{i,t}-\bar{\alpha}_i) + K_{i,t}^c(\beta_{i,t}-\bar{\beta}_i)\big)\right]$,
- approximation errors for $(\hat{\alpha}_i, \hat{\beta}_i)$:
  $\hat{E}_{T,i}^+ = \sum_{t=\tau+1}^T \left[(1-K_{i,t})(\bar{\alpha}_{i,t}-\hat{\alpha}_i) - \frac{1}{2}\big(K_{i,t}(\bar{\alpha}_{i,t}-\hat{\alpha}_i) + (1-K_{i,t})(\bar{\beta}_{i,t}-\hat{\beta}_i)\big)\right]$,
  $\hat{E}_{T,i}^- = \sum_{t=\tau+1}^T \left[K_{i,t}(\bar{\beta}_{i,t}-\hat{\beta}_i) - \frac{1}{2}\big(K_{i,t}(\bar{\alpha}_{i,t}-\hat{\alpha}_i) + (1-K_{i,t})(\bar{\beta}_{i,t}-\hat{\beta}_i)\big)\right]$,
- the deviation of losses caused by the random variables $(K_{i,t})_{i,t}$:
  $L_{T,i}^+ = \sum_{t=\tau+1}^T \left[(1-K_{i,t})\hat{\alpha}_i - \frac{1}{2}(K_{i,t}\hat{\alpha}_i + (1-K_{i,t})\hat{\beta}_i)\right] - (T-\tau)l_i^+$,
  $L_{T,i}^- = \sum_{t=\tau+1}^T \left[(K_i)\hat{\beta}_i - \frac{1}{2}(K_i,t\hat{\alpha}_i + (1-K_{i,t})\hat{\beta}_i)\right] - (T-\tau)l_i^-$,
- the average loss criterion used in `UpdExp`:
  $l_i^+ = l^+(\hat{\alpha}_i, \ \hat{\beta}_i, \ p_i) = (1-p_i)\hat{\alpha}_i - \frac{1}{2}(p_i\hat{\alpha}_i + (1-p_i)\hat{\alpha}_i)$,
  $l_i^- = l^-(\hat{\alpha}_i, \ \hat{\beta}_i, \ p_i) = p_i\hat{\beta}_i - \frac{1}{2}(p_i\hat{\alpha}_i + (1-p_i)\hat{\alpha}_i)$.

We control these terms using the concentration lemmas in Appendix B. Let $\delta > 0$, we define

$$\mathcal{G} = \left\{ \forall i, \ \bar{E}_{T,i}^+ \le 3c\sqrt{2(T-\tau)\log(dT/\delta)} \text{ and } \bar{E}_{T,i}^- \le 3c\sqrt{2(T-\tau)\log(dT/\delta)} \right\},$$

$$\mathcal{H} = \left\{ \begin{array}{ll} \forall i \in [d], \ \forall t > t_{i-1}, & |\bar{\alpha}_i - \hat{\alpha}_{i,t}| \le \sqrt{2\sigma^2 + c^2}\sqrt{2\frac{\log(dT/\delta)+\log(1+4\tau_{i,t})}{\tau_{i,t}+1}} \ ; \\ & |\bar{\beta}_i - \hat{\beta}_{i,t}| \le \sqrt{2\sigma^2 + c^2}\sqrt{2\frac{\log(dT/\delta)+\log(1+4\tau_{i,t})}{\tau_{i,t}+1}} \end{array} \right\},$$

$$\mathcal{I} = \left\{ \begin{array}{ll} \forall i \in [d], & \hat{E}_{T,i}^+ \le (T-\tau)\Big(1 + \frac{\sqrt{2\log(dT/\delta)}}{\sqrt{T-\tau}}\Big)\max\Big(|\bar{\alpha}_i - \hat{\alpha}_i|, |\bar{\beta}_i - \hat{\beta}_i|\Big) \\ & \hat{E}_{T,i}^- \le (T-\tau)\Big(1 + \frac{\sqrt{2\log(dT/\delta)}}{\sqrt{T-\tau}}\Big)\max\Big(|\bar{\alpha}_i - \hat{\alpha}_i|, |\bar{\beta}_i - \hat{\beta}_t|\Big) \end{array} \right\},$$

$$\mathcal{J} = \left\{ \begin{array}{ll} \forall i \in [d], & L_{T,i}^+ \le \frac{3c}{\sqrt{2}}\sqrt{(T-\tau)\log(dT/\delta)}, \\ & L_{T,i}^- \le \frac{3c}{\sqrt{2}}\sqrt{(T-\tau)\log(dT/\delta)} \end{array} \right\}.$$

Applying Lemma 1 and a union bound yields that $\mathbb{P}(\mathcal{G}^c \cup \mathcal{I}^c \cup \mathcal{J}^c) \le \frac{6\delta}{T}$. Likewise Lemma 2 yields $\mathbb{P}(\mathcal{H}^c) \le \frac{4\delta}{T}$.

Besides , $\mathcal{G} \cap \mathcal{H} \cap \mathcal{I} \cap \mathcal{J} \subseteq \mathcal{E}$ (calculations assuming $d \ge 2$, $\tau \le T/2$, and $dT \ge \delta$). ■

### D.2. Proof for the duration of the exploration phase

The following lemma is a consequence of the definition of submodularity, but it is particularly useful when analyzing double-greedy approaches, as it limits the range of possible marginal gains to consider when adding/removing items.

**Lemma 3** *Let $\mathcal{D}$ be a finite set and $f$ be a submodular set-function. Let $A \subset B \subseteq \mathcal{D}$ and an item $i \in (B \setminus A)$. Then,*

$$\Big(f(A \cup \{i\}) - f(A)\Big) + \Big(f(B \setminus \{i\}) - f(B)\Big) \ge 0.$$

We can now use this lemma to prove the following proposition.

**Proposition 3** *For each items $i \in [d]$, under event $\mathcal{E}$ defined in Proposition 2, $\mathtt{UpdExp}$ finds a weight $p_i$ such that $l(\hat{\alpha}_{i,t}, \ \hat{\beta}_{i,t}, \ p_i) + \frac{g_{T,\delta}}{\sqrt{\tau_{i,t}}} \le 0$ before $4\tau_{i,t}$ the current number of exploration rounds for item $i$ has reached the value $4(g_{T,\delta} + \gamma_{T,\delta})^2 \ h_{f,i}$.*

**Proof** We need to look for conditions for Eq. (14) to be satisfied

$$p(\bar{\beta}_i - 3\bar{\alpha}_i) \le -\frac{g_i + \gamma_{T,\delta}}{\sqrt{\tau_i}} + (\beta_i - 2\bar{\alpha}_i), \qquad p(3\bar{\beta}_i - \bar{\alpha}_i) \le -\frac{g_i + \gamma_{T,\delta}}{\sqrt{\tau_i}} + \bar{\beta}_i, \qquad (14)$$

where $\gamma_{T,\delta} = 3\sqrt{(2\sigma^2 + c^2)(\log(dT/\delta) + \log(1 + T))}$.

Considering the different configurations of $(\alpha, \ \beta)$ possible using Lemma 3, gives 5 zones with different sufficient conditions for the existence of a $p_i \in [0, 1]$ satisfying Eq. (14). They are summarized in Table 1, and are upper-bounded by the DG-hardness defined in Definition 2.

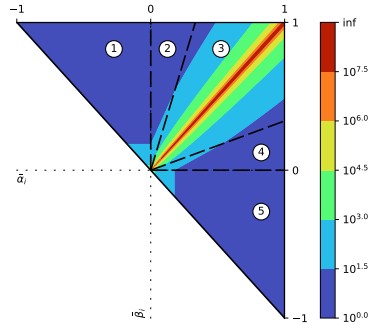

|  | Zone | Threshold of $\frac{\tau_i}{(g_{T,\delta}+\gamma_{T,\delta})^2}$ |
|---|---|---|
| ① | $\bar{\alpha}_i \leq 0, \bar{\beta}_i > 0$ | $1/\bar{\beta}_i^2$ |
| ② | $0 \leq \bar{\alpha}_i \leq \bar{\beta}_i/3$ | $1/(\bar{\beta}_i - 2\bar{\alpha}_i)^2$ |
| ③ | $0 \leq \bar{\beta}_i/3 \leq \bar{\alpha}_i \leq 3\bar{\beta}_i$ | $(\bar{\alpha}_i + \bar{\beta}_i)^2/(\bar{\beta}_i - \bar{\alpha}_i)^4$ |
| ④ | $0 \leq 3\bar{\beta}_i \leq \bar{\alpha}_i$ | $1/(\bar{\alpha}_i - 2\bar{\beta}_i)^2$ |
| ⑤ | $\bar{\alpha}_i > 0, \bar{\beta}_i \leq 0$ | $1/\bar{\alpha}_i^2$ |

Table 1: Exploration thresholds for `UpdExp`.

Figure 3: Exploration thresholds for Subroutine `UpdExp` as a function of $\bar{\alpha}_i$ and $\bar{\beta}_i$ for $c = 1$.