# OpenReview forum: "Logarithmic Regret for Unconstrained Submodular Maximization Stochastic Bandit"
_algorithmiclearningtheory.org/ALT/2025/Conference — ALT 2025_

### Official Review · Reviewer_NEMd · 2024-11-05

**Rating:** 7
**Confidence:** 4

**Review:**

This paper studies online submodular maximization. In particular, it investigates the unconstrained maximization problem, when the underlying function is possibly non-monotone and the feedback received by the learner is bandit. More precisely, at each time step, the learner selects a subset $A_t$ of a $d$ element ground set, is awarded $f(A_t)$, and observes a noisy version of its gain, i.e. perturbed with additive subgaussian noise. Note, the image of $f$ is contained in $[0,c]$ for some $c$.

In the offline problem, it is known that no efficient algorithm can achieve a better-than-0.5 approximation ratio, thus in this paper the natural goal is to achieve sublinear $0.5$ regret.
The contribution of the paper is two-fold:
- On the one hand, they construct the first algorithm with instance-dependent logarithmic regret
- On the other hand, their algorithm is also comparable (although slightly worse) to the state of the art in terms of instance-independent regret bounds.

From the technical point of view, the algorithm is based on the double greedy algorithm by Buchbinder et al. 2012, but entails carefully adapting to the online nature of the problem. In particular, the analysis is non-trivial and interesting.

On the negative side, no lower bound is provided. In particular, it is unclear whether the $T^{2/3}$ dependence on the worst-case regret is necessary. Furthermore, the novelty of the paper is partly hindered by the fact that an algorithm with a slightly better worst-case regret rate is already known (Fourati et al 2023). Finally, to put things in the right perspective, let me mention that the naive algorithm that simply subsample each element of the domain with probability $½$ already provides a $¼$ approximation to the offline problem. Therefore it is immediate to achieve $0$ $¼$-regret.

Overall this is a nice paper, that - in my opinion - clears the bar for ALT.

**Paper Award:**

No

---

> ### Author Response · Authors · 2024-11-22
>
> We thank the reviewer for appreciating our submission, and for valuing its subtleties and its originality.
>
> Unfortunately, we do lose a factor of $\log(d)^{1/3}$ in the worst case compared to the instance-independent bound of Fourati et al. However, as the reviewer noted, this is a slight price to pay for the problem-adaptivity of our algorithm.
>
> We thank the reviewer for the remark about the $1/4$ approximation in the offline problem. It is worth a mention and may be useful for finding lower bounds in future works. It should be noted that, while a zero algorithm yielding a zero $1/4$-regret has no guarantee of yielding a sublinear $1/2$-regret, a sublinear $1/2$-regret does implies a zero $1/4$-regret for big enough $T$.

---

### Official Review · Reviewer_UozH · 2024-11-08
**Review EDITED**

**Rating:** 7
**Confidence:** 2

**Review:**

I am terribly sorry but I initially uploaded the wrong review

Summary

This paper considers the stochastic unconstrained submodular maximisation with bandit feedback problem. This is a sequential setting where in each round a learner chooses an action, which is a set, suffers a loss, which is assumed to be submodular, and subsequently sees the value of the loss at the learner’s action. The goal is to control the 1/2-pseudo regret, which is the difference between half the reward of the best action times the number of rounds and the cumulative rewards of the learner.

The authors provide an algorithm which has logarithmic problem-dependent regret bounds as well as problem-free regret bounds. The problem-dependent regret bounds come with a new notion of hardness for this problem setting.

The algorithm is an explore then commit algorithm. The main technical realisation is that one can use the fact that one can identify the (approximate) best action relatively quickly, after which in the commit phase the authors show that one can guarantee that with high probability, the algorithm no longer suffers any 1/2-regret relative to the best action. Intuitively, the reason for this is that the error in identifying the best action can be compensated for by negative 1/2-reward that is left after bounding the 1-regret.

Strengths and weaknesses

The main results of this paper is a first of its kind for this setting. Prior works only obtained problem-free regret bounds. I like the idea of compensating for the errors in identifying the best action with the negative 1/2-reward that is left after bounding the 1-regret. This could be useful in other settings as well.

The paper is reasonably well written.

The result appears to be correct, although I did not check the proofs in the appendix in depth.

The main thing I found lacking in this paper is a brief discussion of a related, but significantly easier setting: online submodular minimisation with bandit feedback. There have been several works that control the regret through the Lovász extension combined with convex bandit algorithms to obtain T^{2/3} and T^{1/2} regret bounds with efficient algorithms, and a brief discussion of these results is warranted.

**Paper Award:**

No

---

> ### Author Response · Authors · 2024-11-22
>
> We thank the reviewer for their appreciation of our work and for recognizing its originality. We agree that this ``compensation of errors" strategy could have some reach in other settings involving approximation ratios.
>
> > About Submodular Minimization (Online and Bandit)
>
> We did not discuss the minimization settings as they are significantly different and
> much easier (no approximate regret needed) but we agree that the literature deserves a mention and we will add a brief discussion about it in our related work section.

---

### Official Review · Reviewer_F8Nf · 2024-11-10
**Review of "Logarithmic Regret for Unconstrained Submodular Maximization Stochastic Bandit"**

**Rating:** 6
**Confidence:** 3

**Review:**

This paper considers the unconstrained online maximization problem of non-monotonic submodular functions, focusing specifically on situations where only stochastic bandit feedback can be observed. The authors aim to determine the achievable reduction in (1/2)-approximate regret. They propose a natural explore-then-commit algorithm based on the double-greedy algorithm and present an upper bound on the regret achieved by this algorithm. Notably, they define a new metric to measure the problem's difficulty and demonstrate an $ O(\log T) $ regret upper bound dependent on this metric.

**Strengths of the Paper:**
- Prior research is appropriately cited and compared, with a clear explanation of the paper's position in the field. Particularly in Remark 2, the weaknesses of the obtained results are mentioned appropriately.
- The newly proposed concept (a metric for the difficulty of the problem) is explained in a straightforward manner through specific examples.
- The paper demonstrates a steady improvement in the context of online submodular optimization, which is expected to attract interest within this research community.

**Weaknesses of the Paper:**
- The proposed algorithm is a combination of naturally inspired existing techniques (double-greedy and explore-then-commit), and it seems to lack technically non-trivial elements or challenging ideas.
- There is no regret lower bound supporting the algorithm’s effectiveness. However, this is typical in the context of analyzing approximate regret and should not be considered a significant weakness.

**Questions:**
Does the difficulty metric defined in this paper depend on the ordering of the indices of the items in the large set? If so, to what extent does it change depending on the ordering? My intuition suggests that it would be more natural for the intrinsic difficulty of the problem to be independent of the index ordering, and I am curious whether it is possible to define the metric in a way that reflects this.

**Paper Award:**

No

---

> ### Author Response · Authors · 2024-11-22
>
> We thank the reviewer for their review and for acknowledging the clarity and the substantial improvement brought by our submission.
>
> > About the lack of technicality
>
> The base tools that we use (Double-Greedy and ETC) are indeed not new. However, we argue that combining them to get a $\log(T)$ regret bound is *not* straightforward, and needed incorporation of new elements. On the one hand, we introduce the notion of gap and hardness for non-monotone submodular maximization. The latter allows us to define a new stopping criterion for the exploration phase. On the other hand, the exploitation phase requires the idea of committing to weights instead of fixed sets, unlike what is done in previous works.
>
> > About lower bounds
>
> We agree that it would be interesting to have lower bounds and believe it is a promising but non-trivial question for future work.
>
> > Does the difficulty metric defined in this paper depend on the ordering of the indices of the items in the large set?
>
> Yes it does. A permutation-independent metric can be defined by maximizing the one we propose over all permutations. This suggests that the choice of the permutation impacts the performance of the algorithms. Optimizing over it is an interesting question but this problem has been overlooked in the USM literature so far and is beyond the scope of this work.

---

### Official Review · Reviewer_JVqR · 2024-11-11
**Submodular maximization in stochastic bandit setting**

**Rating:** 6
**Confidence:** 3

**Review:**

This paper studies submodular maximization in a stochastic bandit feedback model. There is an unknown submodular function $f$ on some discrete universe U whose output is in the range $[0,1]$. The algorithm at each step picks a set S \subseteq U, and gets reward $f(S) + noise$.   In the offline setting, when $f$ is submodular but not necessarily monotone, and $S$ is unconstrained, there is an algorithm that gets a $1/2$ approximation to the submodular maximization problem. In this work, the authors consider the online stochastic bandit setting, and aims to minimize the gap between the algorithm’s reward and $OPT/2$. This is referred to as the (1/2)-approximate pseudoregret.

The adversarial experts version of this problem has algorithms that achieve an approximation regret of $O(\sqrt{dT})$. In the bandit setting, the work of Niazadeh et al. (2021) achieves a regret bound of $O(dT^{2/3})$. Fourati et al. 2023 achieve regret $O(dT^{2/3} log^{1/3} (dT))$ for the stochastic setting.

The main result in this paper is an adaptation of the Double-Greedy algorithm to this stochastic bandit setting, and shows a 1/2-approximate-regret bound that matches the Fourati et al. result in the worst case and can be better in a problem-dependent setup. In particular, the paper defines a quantity $H_f$ for a submodular function $f$ and shows that the regret of their algorithm is controlled upto logarithmic terms by $H_f$. In particular, this means that the regret is $function(f) \log dT$ and has no polynomial dependence on $T$.

Intuitively, this is achieved by running the Double Greedy Algorithm of Buchbinder-Feldman-Naor-Schwarz, where at each step a
randomized decision is taken. In this algorithm, the randomized decision is made adaptively. Briefly, the new algorithm stops early in this “step” if it finds a $p$ that is guaranteed to be good even accounting for the noise in estimation, it takes that step. Else, it continues until the estimation noise is small enough to lead to low regret. The authors define a quantity depending on the function $f$ (and the ordering of the points ${1,2,…d}$) and show that this quantity allows them to upoer bound the number of steps taken by the algorithm.


*Questions*


- Does the adversarial bandit upper bound of $O(dT^{2/3})$ imply the same bound for the stochastic setting? Why/why not? What does this mean about the Fourati et al. result and your worst-case result?

- The paper is missing a discussion of several relevant references. E.g. Streeter and Golovin: “An Online Algorithm for Maximizing Submodular Functions” seems to be an early work on online submodular maximization, and Yue and Guestrin (Neurips 2011) studies some version of submodular bandits.
Much more relatedly, the current work seems to be much more in the vein on maximizing submodular functions given a noisy oracle. There seems to be a long line of work on this question (e.g. “Noisy Submodular Maximization via Adaptive Sampling with Applications to Crowdsourced Image Collection Summarization” by Singla, Tschiatschek, Krause. Also there is the long line of work on Optimization-from-Samples by Yaron Singer and colleagues, and work on “Stochastic Submodular Optimization”. These seem to me to very intimately related to the current work.

- It would be interesting to understand if this H_f also gives a lower bound on the regret of this algorithm, or better still for any algorithm.

**Paper Award:**

No

---

> ### Author Response · Authors · 2024-11-22
>
> We thank the reviewer for their inputs that we will take into account.
>
> > Does the adversarial bandit upper bound of $O(dT^{2/3})$ imply the same bound for the stochastic setting? Why/why not? What does this mean about the Fourati et al. result and your worst-case result?
>
> Yes indeed, in the stochastic setting, the adversarial algorithm from Niazadeh et al. satisfies a $O(dT^{2/3})$ upper bound (up to $\log(T)$ factors). The result of Fourati et al. matches this upper bound. Our algorithm however improves it to $\log(T)$ by better exploiting the structure of the problem instances, similarly to what happens in the multiarmed bandit setting between adversarial and stochastic algorithms.
>
> In an adversarial setting, both our algorithm and the one from Fourati et al. may incur linear regret, while the one from Niazadeh et al. satisfies a $O(dT^{2/3})$ bound. A best-of-both-worlds type algorithm would be an interesting research direction for future work.
>
> > The paper is missing a discussion of several relevant references. E.g. Streeter and Golovin: “An Online Algorithm for Maximizing Submodular Functions” seems to be an early work on online submodular maximization, and Yue and Guestrin (Neurips 2011) studies some version of submodular bandits. Much more relatedly, the current work seems to be much more in the vein on maximizing submodular functions given a noisy oracle. There seems to be a long line of work on this question (e.g. “Noisy Submodular Maximization via Adaptive Sampling with Applications to Crowdsourced Image Collection Summarization” by Singla, Tschiatschek, Krause. Also there is the long line of work on Optimization-from-Samples by Yaron Singer and colleagues, and work on “Stochastic Submodular Optimization”. These seem to me to very intimately related to the current work.
>
> We have restrained ourselves to the literature which was the closest to our framework: the Unconstrained Submodular Maximization (USM) problem. However, we agree that a more thorough related work section is relevant, and we will strive to achieve it. In particular we will add the references suggested by the reviewer.
>
> Note that we did not include them in our initial submission because they seemed a little too far at first sight. Indeed, they all consider *monotone* submodular functions (as is common in most existing works), for which the known results and techniques differ significantly. Notably, monotone submodular functions allow approximation ratios of $1-1/e$, whereas the non-monotone case restricts this ratio to $1/2$. Moreover, as we understand it, the Optimization-from-Samples literature addresses the problem of optimizing a function given samples from an arbitrary distribution over sets. While this bears some resemblance to off-policy learning, our work operates in a fully online setting.
>
> >It would be interesting to understand if this $H_f$ also gives a lower bound on the regret of this algorithm, or better still for any algorithm.
>
> We agree with that remark and believe it is an interesting but non-trivial question for future work.

---

### Author Rebuttal · Authors · 2024-11-22

We dearly thank the reviewers for their insightful reviews. No major concern was raised and we believe to have addressed the main interrogations. We will broaden our literature review with the suggested pointers.

---

### Meta-Review · Area_Chair_DeY9 · 2024-12-12

**Recommendation:** Accept
**Confidence:** 4

**Metareview:**

The paper studies online submodular maximization in a stochastic bandit setting, specifically for unconstrained, non-monotone submodular functions. The main contributions include developing an algorithm that achieves a ($1/2$)-approximate regret with an $O(\log T)$ bound, that replaces the worst-case polynomial dependence on $T$ with dependence on a novel instance-dependent metric $H_f$ that characterizes the difficulty of the optimization problem for a given submodular function. This approach adapts the Double-Greedy algorithm of Buchbinder et al. (2012) to the stochastic bandit context and improves upon previous works like Niazadeh et al. (2021) and Fourati et al. (2023), matching their worst-case $O(dT^{2/3})$ regret bounds while potentially offering better performance in benign problem instances. The algorithm uses an adaptive approach that can stop early when a promising solution is found, by carefully accounting for the noise in estimation.

Due to these significant contributions and the unanimous support of the reviewers, my recommendation is to accept the paper.  I encourage the authors to carefully consider the suggestions brought up in the reviews, in particular around additional discussion of related work and some missing relevant citations.

**Paper Award:**

No